SCIENTIFIC CORRESPONDENCE

# Comment on 'Parasite defensive limb movements enhance acoustic signal attraction in male little torrent frogs'

**Nigel K Anderson[1], Doris Preininger[2,3], Matthew J Fuxjager[1]\***

[1]Department of Ecology, Evolution, and Organismal Biology, Brown University, Providence, United States; [2]Department of Evolutionary Biology, University of Vienna, Vienna, Austria; [3]Vienna Zoo, Vienna, Austria

**Abstract** Zhao et al. recently reported results which, they claim, suggest that sexual selection produces the multimodal displays seen in little torrent frogs (*Amolops torrentis*) by co-opting limb movements that originally evolved to support parasite defense (Zhao et al., 2022). Here, we explain why we believe this conclusion to be premature.

## Introduction

Many animals communicate by performing multimodal displays that showcase vocal and gestural signals (*Partan and Marler, 1999*; *Bro-Jørgensen, 2010*; *Higham and Hebets, 2013*; *Starnberger et al., 2014b*; *Mitoyen et al., 2019*). Recently, Zhao et al. attempted to study how these displays might evolve, at least with respect to the process by which discrete limb movements can be incorporated into more complex signaling routines. They did this by studying little torrent frogs (*Amolops torrentis*), which inhabit noisy streams throughout Hainan Island in Southern China (*Zhao et al., 2022*). They concluded that: (i) male frogs produce a set of discrete arm and leg maneuvers to help swat away blood-sucking parasites; (ii) these same limb movements enhance the attractiveness of male calls to females. Zhao et al. then argued that natural selection for parasite-induced movements creates an opportunity for sexual selection to generate a multimodal display by integrating these movements into the species' signaling routine. However, we argue that these conclusions are premature because they are based on misinterpretations of the study's main results.

## Results and discussion

### Only "un-preferred" movements are produced around parasites

For the main conclusions of *Zhao et al., 2022* to be correct, the following must be true: (i) limb movements must function to protect frogs from parasitism; (ii) this defense tactic must have emerged *before* the species evolved either its social limb displays or its multimodal communication strategy (*True and Carroll, 2002*; *Borgia and Keagy, 2015*; *Schwark et al., 2022*). However, Zhao et al. do not to provide compelling evidence for either point. For example, they report male frogs sometimes produce certain gestures when parasites land on them *or* when parasites fly in the frog's "vicinity" (although this term is not defined). Moreover, they do not statistically analyze their data to assess whether frogs are more likely to produce gestures when parasites are around. We therefore ran such an analysis, and we found that only two movements —limb shaking (LSA) and wiping (W)—were more likely to occur in the presence of parasites than one might expect by chance (*Figure 1*). Importantly, these specific movements were not the ones that females preferred in choice tests (Figure 5C and 5D in *Zhao et al., 2022*). At the same time, we found that both hind foot lifting (HFL) and arm wiping

**\*For correspondence:**
matthew_fuxjager@brown.edu

**Competing interest:** The authors declare that no competing interests exist.

(AW) were *not* more likely to occur in the presences of parasites (*Figure 1*), even though these were the two limb movements that females seemed to prefer in choice tests (Figure 5A and 5B in *Zhao et al., 2022*). Our results therefore suggest that parasite presence is associated with only certain limb movements that Zhao et al. studied, but none that are positively linked to female preference (but see below for concerns about female preference tests).

## Parasitism and limb movements are correlational, and not causal

Zhao et al. also report a positive correlation between the number of parasite visits males receive and the number of limb movements males produce. They interpret these data as further support for the hypothesis that parasites are the cause of limb movements. However, correlation does not equal causation. Even if males who encountered more parasites were also more likely to have produced limb displays, this relationship does not necessarily mean that parasites directly "induced" or "evoked" this behavior, as Zhao et al. assert. Other explanations for the association include the possibility that higher quality males who display more vigorously also occupy spots along the breeding stream that contain more parasites. Micro- and macro-ecological factors that determine the abundance and distribution of blood-sucking parasites that target frogs are poorly understood (outlined recently by *Virgo et al., 2022*), but other work in midges implies a wide range of factors associated with the local landscape and ecology can influence their distribution and abundance (*Kluiters et al., 2013*; *Rigot et al., 2013*). Alternatively, parasites might be attracted to male calls (*Bernal et al., 2006*; *Aihara et al., 2016*; *Toma et al., 2019*), which males might produce more often when they are using their limbs to display during bouts of male-male competition (*Grafe et al., 2012*). Indeed, in both cases here, we would expect positive correlations between parasite levels and limb movements, without a causal link between the two.

Understandably, one might ask why exactly frogs would evolve limb movements like hind foot lifting (HFL) and arm wiping (AW), if they are not involved in parasite defense. This question seems even more logical given that Zhao et al. classify limbs movements produced in the absence of parasites as "spontaneous," which implies that they are performed at random or without being triggered by an external stimulus. An alternative view, however, is that these so-called "spontaneous" limb movements are actually generated as social signals that help males compete with sexual rivals during agonistic interactions. Most frogs that use gestural signals do so for this purpose (see *Table 1*), and thus the behavior is assumed to evolve through intrasexual selection (*Preininger et al., 2013b*; *Preininger et al., 2013c*; *Mangiamele and Fuxjager, 2018*; *Anderson et al., 2021a*). Zhao et al. do not determine how many of the limb movements produced in the absence of parasites (e.g., "spontaneous") were actually the result of male-male interactions, but they do indicate that little torrent frogs use these movements in such contexts.

## Limitations to the female preference tests

Zhao et al. also conduct experiments that examine whether females prefer to associate with males that produce supposed "parasite-induced" limb movements while calling. In theory, results from this study should provide the rationale for the hypothesis that sexual selection by female choice co-opts leg movements into reproductive displays. Yet, as we indicate above, this idea runs counter to many studies that suggest that gestural displays in frogs mediate agonistic encounters among males (*Table 1*). To our knowledge, there are currently no studies that clearly and definitively show that male frogs use the same limb movements described by Zhao et al. to attract female mates. There is certainly some observational evidence for visual displays employed during courtship, but such data are relatively rare and functionally ambiguous (examples: *de Sá et al., 2018* has n=3 courtship interactions; *Furtado et al., 2019* has n=1 courtship interaction). To this end, Zhao et al. only report four male-female interactions across two breeding seasons, and during these interactions males don't produce any of the limb displays that are purported to be linked with parasite defense. Furthermore, when working in the field with torrent frogs, one must recognize that it is nearly impossible to distinguish male gestural displays directed to other males from those directed at females (see *Table 1* and most "courtship" interactions listed therein). This is because males in the area will trigger these behaviors from each other, even as females approach (Preininger and Fuxjager, personal observations; *Zhao et al., 2022*).

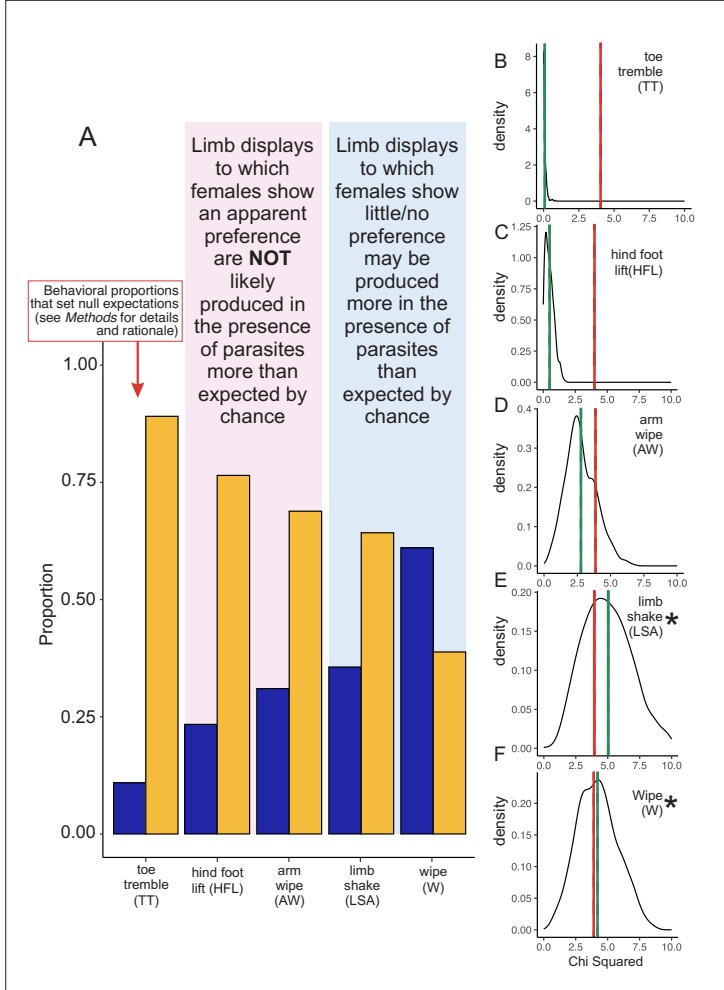

**Figure 1.** Re-analysis of whether male little torrent frogs (*Amolops torrentis*) produce limb displays in the presence of parasites. (**A**) Proportion of different limb displays observed passively in a population of males (n=69) either in the presence of parasites (blue bars) or in the absence of parasites (orange bars). Note that these data are weighted by the number of limb movements each male produced, which were highly skewed in the original dataset. In other words, in the first analysis by Zhao et al., some males produced >90 displays, whereas other males produced zero (*Zhao et al., 2022*). See Methods for details about how we weighted values. For all subsequent analyses (G-tests for goodness of fit), the proportion of toe trembling produced in the presence and absence of parasites was used as the null hypothesis, setting our expectation of how often displays should be produced by chance in the presence or absence of parasites (see *Methods* for justification). (**B–F**) Density plots of the boot strapped chi-squared ($\chi^2$) statistics from the G-test of goodness of fit analysis. On the y-axis is the density of chi-squared ($\chi^2$) statistics after 1,000 iterations, and on the x-axis is the chi-squared ($\chi^2$) value. Solid green lines denote mean chi-squared statistics associated with each distribution of values, whereas solid red lines represent the cut-off for statistical significance ($P<0.05$) with 1 degree of freedom. If the green line falls on the right side of the red line, then the result is statistically significant (i.e., male frogs appear to perform the given display in the presence of parasites more than we might expect by chance, as determined by the null model set through toe trembling). By contrast, if the green line falls on the left side of the red line, then the result is not significant (i.e., male frogs do not perform the given display in the presence of parasites more than we might expect by chance). We found that (**B**) toe trembling (TT) was (as expected) not statistically significant ($\chi^2=0.084$, $P=0.772$), nor was (**C**) hind foot lifting (HFL; $\chi^2=0.487$, $P=0.485$) or (**D**) arm wiping (AW; $\chi^2=2.772$, $P=0.096$). Importantly, these were the behaviors the females supposedly preferred, though see the main text for a discussion of the limitations associated with this assay. We found that (**E**) limb shaking behavior (LSA) was statistically significant ($\chi^2=5.0314$, $P=0.025$, denoted with asterisk), as was (**F**) wiping (W) ($\chi^2=4.212$, $P=0.040$, denoted with asterisk). These latter two behaviors (LSA and W) were not preferred by females in the behavioral assay. Note that when comparing A to both E and F (LSA and W, respectively), the proportions in A would suggest that the effect reported in F would be more robust, compared to the effect in E. However, there were several males that did not wipe (0 values), which may have broadened the Chi Squared curve and decreased the statistical power in the analysis.

Still, Zhao et al. attempt to test female preference for male limb movements by presenting females with video stimuli of males that were calling and either producing limb movements or not. However, these video stimuli are not ecologically relevant to female frogs. This is because each stimulus was manually altered to include a standardized audio channel, such that the male in the video would be perceived to have called without inflating its vocal sac. Free-living females do not naturally encounter such stimuli, particularly when they assess males by looking at them head-on (as females do in this experiment). Zhao et al. indicate that they designed the stimuli this way because they were afraid the effect of vocal sac inflation would mask any effect of limb movement on female preference. Vocal sac inflation has a powerful effect on sexual attractiveness and mate choice in frogs (reviewed by *Starnberger et al., 2014a*), including in little torrent frogs (*Zhao et al., 2021*). Importantly, if vocal sac inflation does mask effects of limb movements on female preference, then selection should not strongly favor the co-option of these movements into the display. We suspect that females showed a preference for males that produced HFL and AW movements because they were the closest resemblance of "fixed" vocal sac inflations, particularly when the alternative stimulus included calls without vocal sac inflations (*Rosenthal et al., 2004*; *Narins et al., 2005*; *Taylor et al., 2008*; *Gomez et al., 2011*; *Preininger et al., 2013a*). Visual and acoustic components might differ in context and dominance, but nevertheless strongly modulate mate choice (*Taylor et al., 2011*). One might argue against our point by saying that females can in fact observe males producing limb movements and calls without seeing vocal sac inflation, such as when females see males from behind. However, such visual perspectives of the male were not incorporated into the experimental design, and thus the current study cannot reveal how females would respond to seeing males perform limb movements from such alternate angles.

## Conclusions

Here, we highlight concerns about a study by Zhao et al. that tried to explain the origins of multimodal display behavior in little torrent frogs (*Zhao et al., 2022*). By reanalyzing data from this study, we show that only certain limb movements are potentially performed more in the presence of parasites, and these are not the movements that females seem to prefer. The study by Zhao et al. also over-interprets correlational evidence to propose that limb movements evolved to avoid parasite attacks. Finally, Zhao et al. cannot determine whether limb movements are functionally significant during male-female interactions because female preference experiments were limited with respect to their ethological relevance.

We also have other concerns about this study. For example, data videos and drawings of limb movements are ambiguous and unclear (e.g., parasites are unclear in Video 1; gesture illustrations in Figure 1E and C show mirror images of the same movements), and there are no data showing how frequently frogs use limb movements to physically wipe away parasites, or whether frogs ever experience parasites in their "vicinity" without producing limb movements. It is also unclear why preference tests were carried out at night, which creates a temporal mismatch with day-recorded video stimuli. Nonetheless, as biologists who study gestural signals in frogs, we remain open to the possibility that visual displays might arise through the co-option of adaptive movements that are unrelated to communication. Similarly, we recognize that the role of female choice in the evolution of frog limb displays is poorly understood and merits further investigation. However, studies exploring these topics should be carried out using approaches that are clear and replicable, so that we can draw lasting conclusions.

## Materials and methods

We used data from the original study (Table S1 in *Zhao et al., 2022*) to statistically test whether male frogs were more likely to produce the various limb movements when parasites were around than one would otherwise expect by chance. We reasoned that this analysis would help us understand whether behaviors that were more closely aligned with parasite presence were also associated with female preference tests. (Please see above for a discussion of the limitations associated with preference tests).

We ran all statistical analyses in R Studio (https://www.rstudio.com), an integrated environment for R 4.13 (https://www.r-project.org). For data preprocessing, we noted that Zhao et al. did not account for the drastic differences in number of behaviors produced by each frog. This oversight can lead to

**Table 1.** List of anuran species that perform limb displays or gestural signals. For Dendrobatoidea, see *Hödl and Amézquita, 2001*. Note that in most cases the term courtship in the *Behavioral Function* column refers to instances in which females make choices about male mates, while males use gestural signals to simultaneously compete.

| Family | Species* | Limb Signals | Sex | Behavioral Function | Evidence | Country of Origin | Ecology | Activity Pattern | Reference |
|---|---|---|---|---|---|---|---|---|---|
| Brachychephalidae | Brachycephalus ephippium | arm waving | M | aggressive, defense | observation | Brazil | forest floor | diurnal | Pombal et al., 1994, Goutte et al., 2017 |
| | B. pitanga | arm waving | M | aggressive, defense | observation | Brazil | forest floor, leaf litter | diurnal | Goutte et al., 2017 |
| Bufonidae | Atelopus limosus | arm waving | M | context not determined | observation | Panama | stream | diurnal | Hödl and Amézquita, 2001 |
| | A. varius | arm waving | M, F | aggressive, to defend sites | observation | Colombia, Costa Rica, Panama | stream | diurnal | Crump, 1988 |
| | A. zeteki | arm waving | M, F | M: agonistic, territorial vigilance, F: intersexual female-male, courtship | experimental/ mirror image | Panama | stream | diurnal | Lindquist and Hetherington, 1996, Lindquist and Hetherington, 1998 |
| | A. chiriquiensis | arm waving | M | call response, amplexus attempt | observation | Costa Rica, Panama | stream | diurnal | Lindquist and Hetherington, 1996, Lindquist and Hetherington, 1998 |
| | | leg-kicking | M | during egg laying in amplexus | observation | | | | Lindquist and Swihart, 1997 |
| Leptodactylidae | Leptodactylus melanotus | foot twitching & back raise | M | aggressive | observation | Central America, Mexico | pond | Diurnal, nocturnal | Brattstrom, 1968, Gregory, 1983 |
| | Crossodactylus gaudichaudii | arm waving | M | conspecific in the vicinity | observation | Brazil | stream | diurnal | Weygoldt and Potsch de Carvalho e Silva, 1992 |
| | | leg stretch | M | aggressive | observation | | | | Weygoldt and Potsch de Carvalho e Silva, 1992 |
| | | leg lift | M | aggressive | observation | | | | Weygoldt and Potsch de Carvalho e Silva, 1992 |
| | C. schmidtii | both legs kicking | M, J* | agonistic, *context not determined | observation | Brazil | stream | diurnal | Caldart et al., 2014 |
| | | leg kicking | M | agonistic | observation | | | | Caldart et al., 2014 |
| | | toe flagging | M, F | agonistic | observation | | | | Caldart et al., 2014 |
| | | toe trembling | M | agonistic | observation | | | | Caldart et al., 2014 |
| | | limb lifting (arm & leg) | M, F, J | agonistic, M: courtship | observation | | | | Caldart et al., 2014 |
| | Hylodes asper | foot flagging | M | agonistic, courtship | observation | Brazil | stream | diurnal | Haddad and Giaretta, 1999, Hartmann et al., 2005 |
| | | toe movement, flagging | M | agonistic | observation | | | | Haddad and Giaretta, 1999, Hartmann et al., 2005 |

*Table 1 continued on next page*

*Table 1 continued*

| Species* | Limb Signals | Sex | Behavioral Function | Evidence | Country of Origin | Ecology | Activity Pattern | Reference |
|---|---|---|---|---|---|---|---|---|
| | leg stretching | M, F | M: agonistic; F: mating | observation, experimental (mirror) | | | | Haddad and Giaretta, 1999, Hartmann et al., 2005 |
| | arm lifting | M | agonistic | observation | | | | Haddad and Giaretta, 1999 |
| | kicking | M | aggressive | observation | | | | Haddad and Giaretta, 1999 |
| | leg lifting | M | agonistic | observation | | | | Hartmann et al., 2005 |
| H. cardosoi | leg stretching (1 leg) | M | advertisement, courtship | observation | Brazil | stream | diurnal | Forti and Castanho, 2012 |
| | leg stretching (2 legs) | M | advertisement, courtship | observation | | | | Forti and Castanho, 2012 |
| | limb lifting | M | advertisement, territorial | observation | | | | Forti and Castanho, 2012 |
| | foot flagging | M, F | advertisement, courtship M: territorial | observation | | | | Forti and Castanho, 2012 |
| | foot flagging +toe wave | M | advertisement, courtship, territorial | observation | | | | Forti and Castanho, 2012 |
| | leg kicking | M | advertisement, courtship, territorial | observation | | | | Forti and Castanho, 2012 |
| H. dayctylocinus | foot flagging | M | agonistic, courtship | observation | Brazil | stream | diurnal | Narvaes and Rodrigues, 2005 |
| | toe wiggling | M | agonistic | observation | | | | Narvaes and Rodrigues, 2005 |
| | leg stretching | M | agonistic | observation | | | | Narvaes and Rodrigues, 2005 |
| | kicking | M | aggressive | observation | | | | Narvaes and Rodrigues, 2005 |
| | arm lifting | M | context not determined | observation | | | | Narvaes and Rodrigues, 2005 |
| H. japi | toe trembling | M | agonistic, advertisement, courtship | observation | Brazil | stream | diurnal | de Sá et al., 2016 |
| | toe flagging | M | agonistic, advertisement, courtship | observation | | | | de Sá et al., 2016 |
| | toes posture | M | agonistic, advertisement, courtship | observation | | | | de Sá et al., 2016 |
| | foot shaking | M | agonistic, advertisement, courtship | observation | | | | de Sá et al., 2016 |
| | leg stretching | M | agonistic | observation | | | | de Sá et al., 2016 |

*Table 1 continued on next page*

*Table 1 continued*

| Species* | Limb Signals | Sex | Behavioral Function | Evidence | Country of Origin | Ecology | Activity Pattern | Reference |
|---|---|---|---|---|---|---|---|---|
| | foot flagging | M | agonistic, advertisement, courtship | observation | | | | *de Sá et al., 2016* |
| | hand shaking | M | agonistic, advertisement, courtship | observation | | | | *de Sá et al., 2016* |
| | arm lifting | M,F | agonistic, courtship | observation | | | | *de Sá et al., 2016* |
| | arm waving | M,F | agonistic, courtship | observation | | | | *de Sá et al., 2016* |
| *H. meridionalis* | toe flagging | M | agonistic | experimental | Brazil | stream | diurnal | *de Sá et al., 2018, Furtado et al., 2019* |
| | toe trembling | M | agonistic | observation | | | | *de Sá et al., 2018, Furtado et al., 2019* |
| | toe posture | M | agonistic | observation, experimental | | | | *de Sá et al., 2018, Furtado et al., 2019* |
| | arm lifting | M, F | M-agonistic, F-reproductive | observation, experimental | | | | *de Sá et al., 2018, Furtado et al., 2019* |
| | arm waving | M, F | M-agonistic & reproductive, F-reproductive | observation, experimental | | | | *de Sá et al., 2018, Furtado et al., 2019* |
| | leg lifting | M, F | M-agonistic & reproductive, F-reproductive | observation, experimental | | | | *Furtado et al., 2019* |
| | foot flagging | M | agonistic | observation, experimental | | | | *Furtado et al., 2019* |
| | foot shaking | M | agonistic | observation | | | | *de Sá et al., 2018* |
| | both legs kicking | F | agonistic | observation | | | | *Furtado et al., 2019* |
| *H. nasus* | toe wiggle | M | agonistic (threat signals) | observation, experimental | Brazil | stream | diurnal | *Weber et al., 2004* |
| | arm waving | M | agonistic (threat signals) | observation, experimental | | | | *Weber et al., 2004* |
| | leg stretch | M | agonistic (threat signals) | observation, experimental | | | | *Weber et al., 2004* |
| *H. phyllodes* | foot flagging | M | agonistic | observation, experimental | Brazil | stream | diurnal | *Hartmann et al., 2005, Augusto-Alves and Toledo, 2021* |
| | leg stretching | M | agonistic, courtship | observation, experimental | | | | *Hartmann et al., 2005* |
| | arm lifting | M | agonistic, advertisement | observation, experimental | | | | *Hartmann et al., 2005, Augusto-Alves and Toledo, 2021* |
| | arm waving | M | context not determined | observation | | | | *Augusto-Alves and Toledo, 2021* |

*Table 1 continued*

| Species* | Limb Signals | Sex | Behavioral Function | Evidence | Country of Origin | Ecology | Activity Pattern | Reference |
|---|---|---|---|---|---|---|---|---|
| | leg lifting | M | agonistic, advertisement | observation, experimental | | | | *Hartmann et al., 2005, Augusto-Alves and Toledo, 2021* |
| | two limbs lifting | M | context not determined | observation | | | | *Augusto-Alves and Toledo, 2021* |
| | toe flagging | M | agonistic | observation, experimental | | | | *Hartmann et al., 2005, Augusto-Alves and Toledo, 2021* |
| | foot shaking | M | context not determined | observation | | | | *Augusto-Alves and Toledo, 2021* |
| | two-leg kicking | M | agonistic | observation | | | | *Augusto-Alves and Toledo, 2021* |
| **Myobatrachidae** | | | | | | | | |
| *Taudactylus eungellensis* | leg stretching | M | context not determined | - | | stream | diurnal | *Hödl and Amézquita, 2001* |
| | foot flagging | M | context not determined | - | | | | *Hödl and Amézquita, 2001* |
| **Hylidae** | | | | | | | | |
| *Boana albomarginata* | Limb lifting | M | agonistic | experimental (mirror) | Brazil | pond margins vegetation | nocturnal | *Hartmann et al., 2005, Furtado and Nomura, 2014* |
| *(Hypsiboas albomarginatus)* | face wiping | M | agonistic | experimental (mirror) | | | | *Furtado and Nomura, 2014* |
| *(Hyla albomarginata)* | toe trembling | M | agonistic | experimental (mirror) | | | | *Hartmann et al., 2005, Furtado and Nomura, 2014* |
| | leg kicking | M | agonistic | experimental (mirror) | | | | *Hartmann et al., 2005, Furtado and Nomura, 2014* |
| *B. raniceps* | limb lifting | M | agonistic | experimental (mirror) | Brazil | ponds or wetlands | nocturnal | *Furtado et al., 2017* |
| *(Hypsiboas raniceps)* | toe/finger trembling | M | agonistic | experimental (mirror) | | stream | nocturnal | *Furtado et al., 2017* |
| *Litoria cooloolensis* | foot flagging | M | agonistic | observation | Australia | tree | nocturnal | *Meyer et al., 2012* |
| *L. genimaculata* | foot flagging | M | agonistic | observation | Australia | stream | nocturnal | *Richards and James, 1992* |
| *L. iris* | leg flicking | M | call response | observation | Papua New Guinea | stream | crepuscular | *Meyer et al., 2012* |
| *L. nannotis* | foot flagging | M | agonistic | observation | Australia | stream | nocturnal | *Richards and James, 1992* |
| | arm waving | M | agonistic | observation | | | nocturnal | *Richards and James, 1992* |
| *L. pearsoniana* | hand waving | M | agonistic | observation | Australia | stream | nocturnal | *Meyer et al., 2012* |
| | leg flicking | M | agonistic | observation | | | | *Meyer et al., 2012* |
| *L. rheocola* | leg stretching | M | agonistic | observation | Australia | stream | nocturnal | *Richards and James, 1992* |

*Table 1 continued on next page*

*Table 1 continued*

| Species* | Limb Signals | Sex | Behavioral Function | Evidence | Country of Origin | Ecology | Activity Pattern | Reference |
|---|---|---|---|---|---|---|---|---|
| | arm waving | M | agonistic | observation | | | | *Richards and James, 1992* |
| *L. fallax* | foot flagging | M | agonistic | observation | Australia | pond | nocturnal | *Meyer et al., 2012* |
| | foot flickering | M | agonistic | observation | | | | *Meyer et al., 2012* |
| | kicking | M | aggressive | observation | | | | *Meyer et al., 2012* |
| *Lysapsus limellum* | Limb lifting | M | agonistic | experimental (mirror) | Brazil | lentic water bodies | nocturnal | *Furtado et al., 2017* |
| *Dendropsophus nanus* | Limb lifting | M | agonistic | experimental (mirror) | Brazil | ponds | nocturnal | *Furtado et al., 2017* |
| *Dendropsophus parviceps* | foot flagging | M | agonistic | observation | Venezuela | streamside ponds | nocturnal | *Amézquita and Hödl, 2004* |
| *Hyla parviceps* | arm waving | M | agonistic | observation | | | | *Amézquita and Hödl, 2004* |
| *Hyla sp. (aff. ehrhardti)* | body wiping (foot) | | courtship | observation | Brazil | forest, bromeliads | nocturnal | *Hartmann et al., 2005* |
| | face wiping (arm) | M, F | courtship | observation | | | | *Hartmann et al., 2005* |
| | foot flagging | M | courtship (far from females) | observation | | | | *Hartmann et al., 2005* |
| | limb lifting (arm + leg) | M | courtship | observation | | | | *Hartmann et al., 2005* |
| *Phyllomedusa boliviana* | foot flagging | M | aggressive | observation | Bolivia | pond | nocturnal | *Jansen and Kohler, 2008* |
| | leg lifting | M | aggressive | observation | | | | *Jansen and Kohler, 2008* |
| | leg stretching | M | aggressive | observation | | | | *Jansen and Kohler, 2008* |
| *P. burmeisteri* | leg stretching | M | agonistic | observation | Brazil | pond | nocturnal | *Abrunhosa and Wogel, 2004* |
| | kicking | M | aggressive | observation | | | | *Abrunhosa and Wogel, 2004* |
| *P. sauvagii* | foot flagging | M | territorial | observation | Argentina, Bolivia, Paraguay, Brazil | pond | nocturnal | *Halloy and Espinoza, 2000* |
| *Scinax eurydice* | leg kicking | M | 2 males far from each other | observation | Brazil | pond (rainy season) | nocturnal | *Hartmann et al., 2005* |
| | limb lifting (arm + leg) | M | 2 males far from each other | observation | | | | *Hartmann et al., 2005* |

*Table 1 continued on next page*

*Table 1 continued*

| | Species* | Limb Signals | Sex | Behavioral Function | Evidence | Country of Origin | Ecology | Activity Pattern | Reference |
|---|---|---|---|---|---|---|---|---|---|
| Centrolenidae | Vitreorana uranoscopa (Hyalinobatrachium uranoscopum) | limb lifting (arm +leg) | M | agonistic, spontaneous (no other individual present) | observation | Brazil | | nocturnal | *Hartmann et al., 2005* |
| Ranidae | Pulchrana (Rana) baramica | toe waving | M | attract prey | observation | Singapore | forest | | *Grafe, 2008* |
| | Staurois latopalmatus | arm waving | M | agonistic | observation | Borneo | stream | diurnal | *Preininger et al., 2009* |
| | | foot flagging | M | agonistic | observation | | | | *Preininger et al., 2009* |
| | S. guttatus | foot flagging | M, F | agonistic | F-experimental, M-observation | | stream | | *Grafe and Wanger, 2007, Preininger et al., 2016* |
| | | leg drumming | M | context not determined | observation | | | | *Grafe and Wanger, 2007* |
| | | foot raising | M | courtship | observation | | | | *Grafe and Wanger, 2007* |
| | | arm waving | M | agonistic | observation | | | | *Grafe and Wanger, 2007* |
| | S. parvus | foot flagging | M, J | agonistic | observation, experimental | Borneo | stream | diurnal | *Grafe et al., 2012, Preininger et al., 2012, Preininger et al., 2013b* |
| | | foot lifting (tap) | M | agonistic | observation, experimental | | | | *Grafe et al., 2012, Preininger et al., 2013b* |
| Micrixalidae | Micrixalus candidus | foot lifting | M | agonistic | observation | India | stream | diurnal | Preininger and Fuxjager, pers. observation |
| | | foot stretching | M | agonistic | observation | | | | Preininger and Fuxjager, pers. observation |
| | | foot flagging | M | agonistic | observation | | | | Preininger and Fuxjager, pers. observation |
| | M. elegans | foot lifting | M | agonistic | observation | India | stream | diurnal | Preininger and Fuxjager, pers. observation |
| | | foot stretching | M | agonistic | observation | | | | Preininger and Fuxjager, pers. observation |
| | | foot flagging | M | agonistic | observation | | | | Preininger and Fuxjager, pers. observation |
| | M. kottigeharensis | foot lifting | M | agonistic | observation | India | stream | diurnal | *Preininger et al., 2013c, Anderson et al., 2021b, Anderson et al., 2021d* |
| | | foot stretching | M | agonistic | observation | | | | *Preininger et al., 2013b, Preininger et al., 2013c* |
| | | foot flagging | M | agonistic | observation | | | | Preininger and Fuxjager, pers. observation |

*Table 1 continued on next page*

*Table 1 continued*

| Species* | Limb Signals | Sex | Behavioral Function | Evidence | Country of Origin | Ecology | Activity Pattern | Reference |
|---|---|---|---|---|---|---|---|---|
| | toe wiggling | M | agonistic | observation | | | | Preininger and Fuxjager, pers. observation |
| | kicking | M | aggressive | observation | | | | **Preininger et al., 2013c** |
| M. niluvasei | foot lifting | M | agonistic | observation | India | stream | | **Anderson et al., 2021d,** Preininger and Fuxjager, pers. observation |
| | foot stretching | M | agonistic | observation | | | | Preininger and Fuxjager, pers. observation |
| | foot flagging | M | agonistic | observation | | | | Preininger and Fuxjager, pers. observation |
| | kicking | M | aggressive | observation | | | | Preininger and Fuxjager, pers. observation |
| M. saxicola | foot lifting | M | agonistic | observation | India | stream | | **Anderson et al., 2021d,** Preininger and Fuxjager, pers. observation |
| | foot stretching | M | agonistic | observation | | | | Preininger and Fuxjager, pers. observation |
| | foot flagging | M | agonistic | observation | | | | Preininger and Fuxjager, pers. observation |
| | toe wiggling | M | agonistic | observation | | | | Preininger and Fuxjager, pers. observation |
| | kicking | M | aggressive | observation | | | | Preininger and Fuxjager, pers. observation |
| M. specca | foot lifting | M | agonistic | observation | India | stream | | Preininger and Fuxjager, pers. observation |
| | foot flagging | M | agonistic | observation | | | | Preininger and Fuxjager, pers. observation |
| M. uttaraghati | foot lifting | M | agonistic | observation | India | stream | | Preininger and Fuxjager, pers. observation |
| | foot stretching | M | agonistic | observation | | | | Preininger and Fuxjager, pers. observation |
| | foot flagging | M | agonistic | observation | | | | Preininger and Fuxjager, pers. observation |
| | toe wiggling | M | agonistic | observation | | | | Preininger and Fuxjager, pers. observation |

*Table 1 continued on next page*

*Table 1 continued*

| | Species* | Limb Signals | Sex | Behavioral Function | Evidence | Country of Origin | Ecology | Activity Pattern | Reference |
|---|---|---|---|---|---|---|---|---|---|
| Rhacophoridae | *Buergeria japonica* | leg-stretches | M | agonistic male-male interaction | observation | Japan | aquatic and terrestrial | | *Anderson et al., 2021c* |
| | *B. otai* | foot-flagging | M | agonistic male-male interaction | observation | Taiwan | stream | | *Yang, 2022* |
| | *Theloderma bambusicolum* | foot-flagging | M | territorial behavior | observation | Vietnam | dense bushes | | *Orlov et al., 2012* |

*Species names in parentheses represent former names used in original publication.
M=male. F=female. J=juvenile.

certain individuals in the population having an outsized effect on statistical outcomes. For example, a frog that produced ≈90 limb movements in the absence of parasites and 10 limb movements in the presence of parasites was compared to another frog that produced 10 limb movements in absence of parasites and 1 limb movement in the presence of parasites. The proportion of behaviors that these individuals produced in each context is the same, but the absolute total number of these behaviors is quite different; as a result, if raw values of behavior are compared between the groups (absence of parasites vs. presence of parasites), then the first frog will have a more robust impact than the second frog. Weighting values can be an important way to avoid such effects, and so we adopted this approach. We weighted following *Garamszegi, 2014*, where each display count, X, was multiplied by the inverse of the sum count of X for the given individual.

Next, to test how limb movements might correspond to the presence of parasites, we used a G-test (for goodness of fit) to statistically compare the proportion of limb movements produced in the absence of parasites (i.e., called "spontaneous" limb movements, see main text) and the proportion of limb movements produced in the presence of parasites. This test assumes independence between the proportions. To meet this assumption, we randomly sampled 35 individuals from the data set, and noted the total number of "spontaneous" limb movements these individuals produced. We then took the remaining 34 individuals from the data set and recorded only the total number of limb movements produced in the presence of parasites. We repeated this process 1,000 times, always resampling the dataset with replacement. In each case, we employed the *g.test* function from the *AMR* package to calculate a Chi Squared ($\chi^2$) test statistic, which produced a distribution of statistic values. We used the mean $\chi^2$ statistic associated with each limb movement to compute a corresponding *p* value. Importantly, these models were calculated using a null distribution that was determined by the level of toe trembling behavior in the absence (89%) and presence of parasites (11%). Past studies, including some that Zhao et al. cite (such as *Hödl and Amézquita, 2001*), show that toe trembling is not a parasite defense behavior; rather, it is commonly used either as a social signal (*Lindquist and Hetherington, 1996*; *Rojas and Pašukonis, 2019*) or as a feeding/hunger signal (*Grafe, 2008*; *Hagman and Shine, 2008*; *Sloggett and Zeilstra, 2008*; *McFadden et al., 2010*; *Claessens et al., 2020*). Either way, toe trembling provides a nice statistical heuristic to anchor our a priori expectations of how many of these limb displays should be produced when parasites are not around vs. when they are around. Accordingly, if the proportion of limb displays differed significantly from this expectation, then we could conclude that the given behavior was produced more often in the presence of parasites than expected by chance. By contrast, if the proportion of limb displays *did not* differ significantly from our null expectation based on toe trembling, then we cannot reject the null hypothesis.

## Acknowledgements

We thank Nick Antonson, Nicole Moody, and Sofia Piggott for helpful discussions about this paper.

## Additional information

### Funding

| Funder | Grant reference number | Author |
| --- | --- | --- |
| National Science Foundation | OISE-1952542 | Matthew J Fuxjager |
| Vienna Zoo | | Doris Preininger |

The funders had no role in study design, data collection and interpretation, or the decision to submit the work for publication.

### Author contributions

Nigel K Anderson, Conceptualization, Data curation, Formal analysis, Validation, Investigation, Visualization, Methodology, Writing – original draft; Doris Preininger, Conceptualization, Data curation, Formal analysis, Supervision, Validation, Investigation, Methodology, Writing – original draft, Project

administration; Matthew J Fuxjager, Conceptualization, Formal analysis, Supervision, Funding acquisition, Investigation, Visualization, Methodology, Writing – original draft, Project administration

### Author ORCIDs
Nigel K Anderson ⬤ https://orcid.org/0000-0003-2619-3405
Doris Preininger ⬤ http://orcid.org/0000-0001-6842-1133
Matthew J Fuxjager ⬤ http://orcid.org/0000-0003-0591-6854

### Decision letter and Author response
Decision letter https://doi.org/10.7554/eLife.89134.sa1

## Additional files

### Supplementary files
• MDAR checklist

### Data availability
Figure 1 source data are included with original manuscript (Supplementary file 1) on which we are commenting.

The following previously published dataset was used:

| Author(s) | Year | Dataset title | Dataset URL | Database and Identifier |
|---|---|---|---|---|
| Zhao L | 2022 | The data of parasite-induced and spontaneous displays in each limb movement for calling males, silent males and males that have females nearby | https://doi.org/10.5061/dryad.f1vhhmgzg | Dryad Digital Repository, 10.5061/dryad.f1vhhmgzg |

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
