## [Decision Letter]

Since no essential revisions were requested there is no accompanying author response.

Thank you for submitting your article Comment on ‘Parasite defensive limb movements enhance acoustic signal attraction in male little torrent frogs’ to eLife for consideration as Scientific Correspondence.

Your article, and the response to your comment by Cui and colleagues, have been reviewed by two peer reviewers (who have opted to remain anonymous), and the evaluation was overseen by Ammie Kalan as the Reviewing Editor and Christian Rutz as the Senior Editor.

We have decided to accept for publication both your original comment and the response to it. The substantive comments in this decision letter will also be published as part of your article (and likewise for the comment from Cui).

*Reviewer #1:*

This is a comment on an article that argued that movements made by male frogs to deter parasites were co-opted into a signal that functions in female attraction. The authors of the comment argue that these conclusions are premature and potentially erroneous and present a re-analysis of the data suggesting that there isn't a strong relationship between those movements preferred by females and their use in anti-predator behavior.

In general I agree with the comment, I think that although the idea of parasite movements as signal precursors is very interesting, the data presented in the original manuscript don't provide enough information to test that hypothesis. I also agree with the concerns about how the data were analyzed, and that there is not yet enough information to show that parasites in fact cause the limb movements. That said, I think the argument about limb displays being primarily used in male-male interactions is overstated and is a weak critique at best of the original paper. While it is certainly true that most published evidence is about limb displays as signals to other males, as pointed out by the original authors that does not provide evidence that they aren't also used to signal to females; the lack of evidence for females could be due to something as simple as bias in the topics under study or the greater difficulty of experiments with females in these species. Several of the other arguments made in this comment are much more convincing (e.g. the issues associated with video playbacks, the analyses of when limb movements occurred, and in particular the lack of evidence for the evolutionary scenario implied by the authors of the original study, which I know came up in the correspondence but could be highlighted more in this comment).

Specific comments:

First paragraph in the “Only “un-preferred” movements are produced around parasites” subsection: another possibility maybe worth mentioning is that the parasites could be attracted directly to the limb displays, such that males that give more limb displays end up with more parasites nearby.

Second paragraph in the “Limitations to the female preference tests” subsection: I agree with the concerns in this paragraph, although to me the bigger concern is that there wasn't really a control video. We know from previous studies of video playbacks that frogs are attracted to the light from the monitor regardless of what's on the screen, and also to moving objects that don't resemble frogs (e.g. cubes). So the design of the female preference experiment makes it difficult to interpret the results as favoring female preference for these kinds of movements, because there weren't any controls (which could have been something like other moving objects on the screen, or simply something that wasn't a frog picture on there). That may be worth mentioning, and citing studies that have discussed the problems associated with video playbacks in frogs.

*Reviewer #2:*

- does the Comment challenge one or more of the findings in the original paper in a way that is convincing enough to merit publication?

Yes

The authors of the Comment make some good points here. These frogs have a variety of limb movements but in the reanalysis of the data by these authors show that only two of the movements, hind foot lifting and arm wiping, are associated with the presence of parasites. But these little movements are not the ones that enhance the attraction of the calling male to female.

The second point argues that the authors have demonstrated correlation but not causation. I agree with this, it does seem logical that if a fly lands on the frog and they swipe it away with wiping behavior, as opposed to hindlimb movements, it seems logical to suggest that the limb movement was in response to the parasites landing. True they are not demonstrating cause-and-effect, so perhaps this suggests that the author should be a little more circumscribed and discuss in more detail the fact that the data are correlation.

The authors also criticize the female choice tests. I do share their concern that the authors eliminated vocal sac inflations in their playbacks.

- if yes, are any revisions required before the Comment can be accepted for publication.

No